# Improved resistance of neural networks to adversarial images through generative pre-training

## Abstract

We train a feed forward neural network with increased robustness against adversarial attacks compared to conventional training approaches. This is achieved using a novel pre-trained building block based on a mean field description of a Boltzmann machine. On the MNIST dataset the method achieves strong adversarial resistance without data augmentation or adversarial training. We show that the increased adversarial resistance is correlated with the generative performance of the underlying Boltzmann machine.

## 1 Introduction

In an adversarial attack on a classification system one seeks inputs that lie very close (according to some norm or measure) to a regular input, but give a completely different classification result. Adversarial attacks on machine learning systems have been studied for over a decade, starting from the study of attacks on spam filtering systems (for a historical review see Biggio and Roli (2018)). Adversarial attacks on deep learning image recognition models have first been discussed by Szegedy et al. (2013). In addition to highlighting security implications adversarial images have acquired a new meaning in this context, presenting us with a fundamental conundrum: How can deep learning systems succesfully generalise and at the same time be extremely vulnerable to minute changes in the input? The topic of adversarial images in neural nets has been studied intensely with attacks and defences leapfrogging each other (for an overview see Kurakin et al. (2018)). All common neural network models suffer from susceptibility to adversarial attacks. It has even been shown by Moosavi-Dezfooli et al. (2017) that adversarial attack patterns can be transferred between different models, suggesting some common underlying mechanism. Despite the intense study of the subject a solution remains elusive, with Jetley et al. (2018) suggesting that classification performance and vulnerability to adversarial attacks are correlated, while Gilmer et al. (2018) present evidence for the opposite in a simple model. The difficulty in finding neural networks robust against adversarial attacks suggest that the standard discriminative neural network training procedure and the resulting networks' vulnerability are linked. Making use of generative methods (e.g variational autoencoders, introduced in this context by Gu and Rigazio (2014)) has resulted in creating the most adversarially robust classifications but with added computational cost at inference time (Schott et al. (2018)). Another state of the art approach to increasing adversarial resistance applies denoising to image patches to achieve strong robustness (Moosavi-Dezfooli et al. (2018)).

In this article we concentrate on one common generative model, Boltzmann machines, and the impact of its use on adversarial resistance. We examine the adversarial resistance on the standard MNIST dataset. Boltzmann machines are a stochastic model with binary units (see e.g. Hinton (2007)). A number of "hidden" units form a model for the data distribution over the "visible" units. In a classification scenario one would fix the visible units and use the probability distribution over the hidden units for further processing. In a restricted Boltzmann machine the hidden units are independent and therefore the distribution over the hidden units in a classification scenario becomes a product distribution fully characterised by the average values of the individual hidden units. Pre-training using restricted Boltzmann machines has been used in early neural networks (Hinton et al. (2006)) but was abandoned as it became clear that good training can be achieved without it. Here we follow a slightly different path. We will use a Boltzmann machine to model image patches only. We do the modelling in a hierarchical fashion: train one Boltzmann machine for small image patches,

then add more hidden units to model the joint distribution over larger patches, similar to the stacked RBM approach of Norouzi et al. (2009). This forces us to depart from a simple restricted Boltzmann machine model and therefore we cannot obtain the hidden probability distribution as easily as for a restricted Boltzmann machine. Extended mean field methods developed for the description of spin glasses in physics (Thouless et al. (1977); Georges and Yedidia (1991); Kühn and Helias (2017)) have been applied succesfully to train a restricted Boltzmann machine by Gabrié et al. (2015) and Budzianowski (2016). We use these methods to arrive at a description of the hidden units in terms of their average values, where now we replace the single neural network layer resulting from the restricted Boltzmann machine with a many-layer iteration derived from the mean field description. This pre-trained building block can be used as a fixed input layer in a neural network trained in the usual discriminatory fashion.

In summary the contributions of this article are as follows:

- We construct a new neural network building block, based on the mean field description of a Boltzmann machine. The result is a deep feed forward neural network with shared weights and a structure derived from the energy function of the underlying Boltzmann machine.

- We use this building block in a neural network to achieve state of the art resistance to adversarial attacks on MNIST and show a correlation between the loss function of the Boltzmann machine (i.e. it's generative capability) and the adversarial resistance of the resulting neural network.

- We show that the function mapping visible unit inputs to hidden unit outputs implemented by the new building block is strongly dependent on the input. A ball of random distortions around a training example gets mapped to a smaller output volume compared to the same mapping starting from a random input.

## 2 Boltzmann Machines as Feature Extractors

Let us review the the mean field description of a Boltzmann machine (Budzianowski (2016); Gabrié et al. (2015)). Boltzmann machines are stochastic, energy based models derived from the statistical physics of interacting spins. Boltzmann machines are a generative model, the generated probability distribution being the thermal equilibrium probability distribution over all possible spin combinations $\boldsymbol{s} = (s_1, s_2, \ldots, s_N)$ with $s_n = \pm 1$, as determined by the energy function through its parameters $\boldsymbol{\theta} = (\boldsymbol{b}, \boldsymbol{J})$

$$E(\boldsymbol{\theta}, \boldsymbol{s}) = \boldsymbol{s} \cdot \boldsymbol{J} \cdot \boldsymbol{s} + \boldsymbol{b} \cdot \boldsymbol{s},$$

so that the probability of realising the configuration $\boldsymbol{s}$ is proportional to the Boltzmann factor

$$p(\boldsymbol{\theta}, \boldsymbol{s}) \propto e^{-E(\boldsymbol{\theta}, \boldsymbol{s})}.$$

The spins are divided into two sets, the visible units $\boldsymbol{v}$ and the hidden units $\boldsymbol{h}$. When training a Boltzmann machine we want to generate a target probability distribution $p_0(\boldsymbol{v})$ over the visible units, as defined by example configurations $\boldsymbol{v}_j^{(0)}$, $j = 1, \ldots, n_J$. This can be achieved by adjusting the parameters of the energy function to minimise the relative entropy between the marginal distribution over the visible units

$$p(\boldsymbol{v}, \boldsymbol{\theta}) = \mathrm{Tr}_{\boldsymbol{h}} \, p(\boldsymbol{v}, \boldsymbol{h}, \boldsymbol{\theta}).$$

and the target distribution:

$$D(p_0||p) = \sum_{j=1}^{n_J} p_0(\boldsymbol{v}_j^{(0)}) \log p_0(\boldsymbol{v}_j^{(0)}) - p_0(\boldsymbol{v}_j^{(0)}) \log p(\theta, \boldsymbol{v}_j^{(0)}). \tag{1}$$

The gradient of the relative entropy with respect to the parameters of the energy function $\theta$ can be expressed in terms of expectation values of spins

$$\frac{dD(p_0||p)}{db_k} = \sum_{j=1}^{n_J} \frac{p_0(\boldsymbol{v}_j^{(0)})}{p(\theta, \boldsymbol{v}_j^{(0)})} \langle s_k \rangle_{C(j)} - \langle s_k \rangle_F \,, \tag{2}$$

and spin-spin correlators

$$\frac{dD(p_0||p)}{dJ_{kl}} = \sum_{j=1}^{n_J} \frac{p_0(\boldsymbol{v}_j^{(0)})}{p(\theta, \boldsymbol{v}_j^{(0)})} \langle s_k s_l \rangle_{C(j)} - \langle s_k s_l \rangle_F \,, \tag{3}$$

where $\langle \ldots \rangle_F$ denotes the expectation value of the model without constraints (free) and $\langle \ldots \rangle_{C(j)}$ the expectation value if the visible units are fixed to example $\boldsymbol{v}_j$ (clamped). This gradient can be used in a first order optimisation method to arrive at an optimal parameter configuration $\theta_0$. While the expression for the gradient looks simple, evaluating the probabilities and expectation values is not, and some approximation scheme needs to be used. Two schemes are usually used, one based on sampling (Hinton and E. (2002)), the other one based on a mean field approximation. Here we will concentrate on the mean field approximation.

The central quantity in mean field theory is the free energy. We can obtain the free energy of our model as an expansion in the order of the coupling $J$ as derived by a number of authors (Georges and Yedidia (1991); Kühn and Helias (2017)).

$$\mathcal{F} = \sum_n F_n$$

To lowest order the free energy is given by the entropy of independent spins.

$$F_0 = \sum_{i=1}^N \left(\frac{1-m_i}{2}\right) \log\left(\frac{1-m_i}{2}\right) + \left(\frac{1+m_i}{2}\right) \log\left(\frac{1+m_i}{2}\right),$$

where $m_i$ is the magnetisation of spin $i$. Up to second order we obtain

$$F_1 = \sum_{\langle ij \rangle} J_{ij} m_i m_j + \sum_i b_i m_i,$$

$$F_2 = \sum_{\langle ij \rangle} J_{ij}^2 \left(1 - m_i^2\right) \left(1 - m_j^2\right),$$

where the notation $\langle ij \rangle$ denotes summation over unequal indices. For orders up to four see appendix A. From the free energy we can derive an equation to obtain the mean field values $m_i$ in the approximation up to order $N$

$$m_i = \tanh\left(\sum_{n=1}^N R_i^{(n)}\right) \tag{4}$$

with

$$R_i^{(n)} = \frac{dF_n}{dm_i}.$$

In particular, the first two orders are

$$R_i^{(1)} = b_i + 2 \sum_j J_{ij} m_j$$

and

$$R_i^{(2)} = -4m_i \sum_j J_{ij}^2 (1 - m_j^2).$$

Higher orders can be found in the appendix A. In the free case, the transcendental equation, Eq. 4 can be solved by iteration, starting from zero or random magnetisations, until the magnetisations converge. In the clamped case we keep the visible magnetisations fixed, while iterating over the hidden units. This results in two sets of distinct magnetisations: the free magnetisations $m_F$ and the clamped magnetisations $m_{C(j)}$, where example $j$ is fixed on the visible units. We also get a set of corresponding free energies $\mathcal{F}_F$ and $\mathcal{F}_{C(j)}$. The correlation functions can be obtained from the free energies

$$\langle s_k s_l \rangle_{F/C(j)} = \frac{1}{2} \frac{d\mathcal{F}_{F/C(j)}}{dJ_{kl}},$$

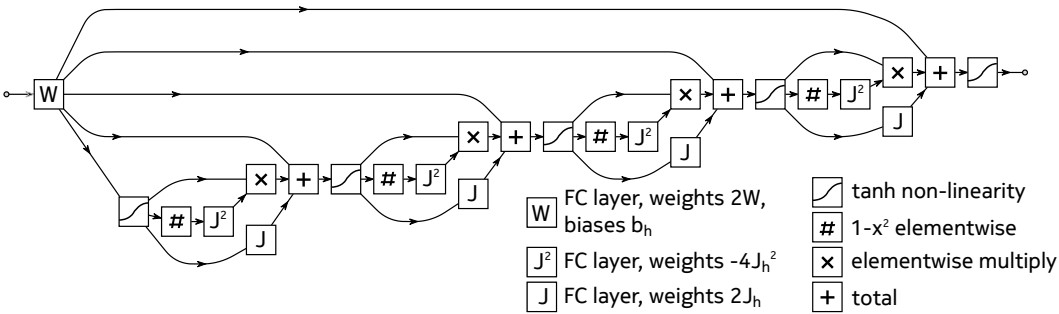

Figure 1: An example of an unrolling of the second order mean field iteration into a 5 layer neural network. The weights in all layers except the first one are shared, resulting in a deep network with a small number of parameters. The structure of the network is fully determined by the mean field equations derived from the energy function of Boltzmann machine.

and the expectation values are given directly by

$$\langle s_k \rangle_{F/C(j)} = m_{F/C(j),k}.$$

These expectation values can be inserted in the expressions for the gradient, Eqs. 2 and 3, and used to minimise the relative entropy Eq. 1. Probabilities can be estimated from the free energies

$$p(\theta, \boldsymbol{v}_j^{(0)}) = e^{\mathcal{F}_{C(j)} - \mathcal{F}_F}.$$

The training of a Boltzmann machine emphasises the main task of the machine as generating examples from the target distribution, but one can look at the machine from a different point of view: using the hidden units for constructing a model of the example data. The probability distribution over the hidden units if an example is fixed on the visible units then tells us something about the model required to describe the example. In a restricted Boltzmann machine (RBM) the probability distribution is a product distribution over the hidden units and features, but one can imagine introducing couplings between hidden units to account for correlations between individual features to arrive at a better/simpler description of the data. In the remaining article we will discuss machines that have bipartite connectivity between visible and hidden units and full connectivity between hidden units.

During inference we want to obtain the hidden magnetisations given the magnetisations of the visible units. We split the magnetisations into visible and hidden parts $\boldsymbol{m} = (\boldsymbol{m}_v, \boldsymbol{m}_h)$. Similarly we split the magnetic fields $\boldsymbol{b} = (0, \boldsymbol{b}_h)$ and the couplings

$$\boldsymbol{J} = \left( \begin{array}{cc} 0 & \boldsymbol{W} \\ \boldsymbol{W} & \boldsymbol{J}_h \end{array} \right),$$

where $\boldsymbol{W}$ now describes the bipartite connectivity between visible and hidden units and $\boldsymbol{J}_h$ the coupling between hidden units. Keeping terms to second order the mean field equations become

$$\boldsymbol{m}_h = \tanh \left[ 2\boldsymbol{W} \cdot \boldsymbol{m}_v + \boldsymbol{b}_h + 2\boldsymbol{J}_h \cdot \boldsymbol{m}_h - 4\boldsymbol{m}_h \boldsymbol{J}_h^2 \cdot \left( 1 - \boldsymbol{m}_h^2 \right) \right]. \tag{5}$$

Solving the mean field equation Eq. 5 requires iteration, e.g. starting from the initial magnetisations $\boldsymbol{m}_h = \boldsymbol{0}$. The iteration can be unrolled into an $r$ layer neural network, where $r$ is greater than the number of iterations necessary for convergence (see Fig. 1). We will refer to this neural network as the mean field Boltzmann network.

## 3 TRAINING THE MEAN FIELD BOLTZMANN NETWORK

The Boltzmann machine underlying the mean field Boltzmann network is trained on data derived from the standard MNIST training set. We binarize the MNIST images with a threshold (50/255) and create two datasets. The first dataset contains 4 x 4 patches cut from the binarised images, the second contains 8 x 8 patches. We start by training a Boltzmann machine for the 4 x 4 patches. Of all $2^{16}$ possible patches, 679 patches contain over 98% of all occurences in the dataset. We use these 679 to train a mean field Boltzmann machine with 32 hidden units (see Fig. 2). Four of

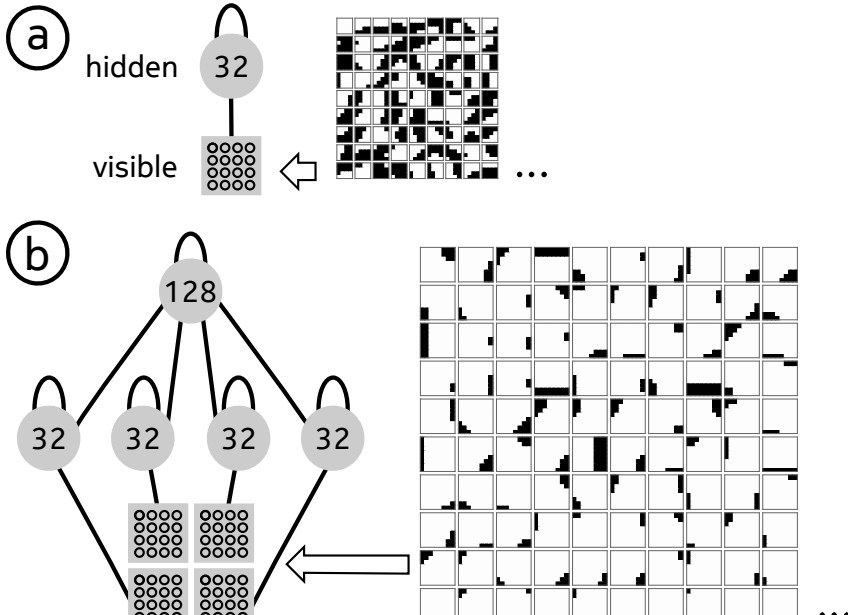

Figure 2: Illustration of the structure of the Boltzmann machine and the training procedure. a) shows the Boltzmann machine used in the first training step consisting of 16 visible units and 32 hidden units. The line between the visible and hidden units represents bipartite connectivity between the units in the two sets. The loop on top of the hidden units indicates full connectivity between hidden units. 4 x 4 patches from the example data are used to generatively train the machine. b) shows the structure of the Boltzmann machine used in the second training step, consisting of 64 visible units and 256 hidden units. The trained small machines are assigned a 4 x 4 sub-patch in the 8 x 8 patches. Each of the 32 hidden units in the small machines gains bipartite connectivity to a further fully connected 128 hidden units and the full machine is trained according to the standard Boltzmann machine training procedure.

these trained small machines are then combined with 128 hidden units to model the 8 x 8 patches. The dataset of 8 x 8 patches contains approximately 600000 examples. For the calculation of the gradient we use batches of 10000 examples. To calculate the free expectation values we use the mean field expansion up to fourth order, for the clamped expectation values it suffices to go to second order. The magnetisations in the free case are calculated using 100 mean-field iterations, for the clamped magnetisation we use 20 iterations. In the late stage of the training the value for the free partition function is underestimated. This can lead to the sum of individual sample probabilities exceeding one. We adjust the free partition function in calculation of the relative entropy and the gradient so that the probabilities within one batch are normalised to one. Similar results to the staged training can be obtained by training a Boltzmann machine of identical connectivity directly on the 8x8 patches.

## 4 ADVERSARIALLY ROBUST NEURAL NETWORKS

The trained mean field Boltzmann network constitutes a new neural network building block that takes in 8 x 8 patches and outputs a vector of 128 hidden magnetisations. This building block has identical input/output dimensions to a standard convolutional layer with 8 x 8 filter size and 128 filters. We can use this building block as an input layer in an identical fashion to a standard convolutional layer, sweeping it over the two input image dimensions. Starting from the 28 x 28 MNIST images, this results in an intermediate representation of 128 x 21 x 21 dimensions. On top of this representation we build a standard three layer convolutional neural network with ReLU activations. We add 3 convolution layers (kernel size 3 x 3, stride 2) with 128 filters, 64 filters and 32 filters, respectively, followed by a fully connected layer and a softmax function). The input

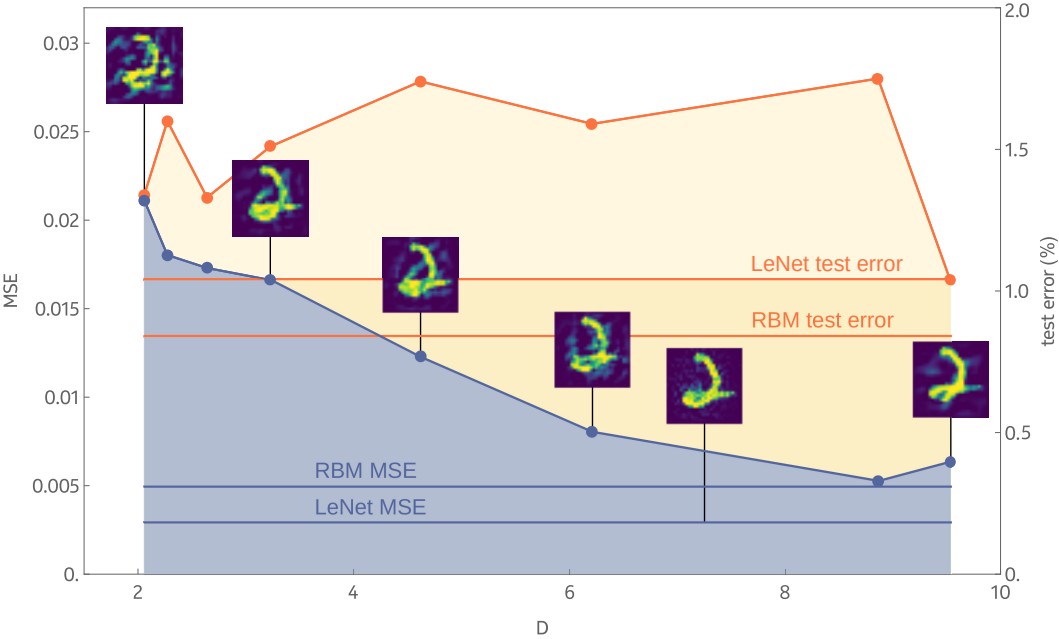

Figure 3: Dependence of adversarial resistance and test error on Boltzmann machine pre-training for a neural network with a mean field Boltzmann network input layer. The blue plot shows the mean square error (for image pixels in the range $[0, 1]$) required to create an adversarial image averaged over 1000 random samples from the MNIST test set as a function of the relative entropy $D$ achieved in Boltzmann machine pre-training. Lower relative entropy at the pre-training stage results in higher resistance to adversarial attacks. The blue lines indicate the adversarial resistance for LeNet and a pre-trained first RBM layer. Attached to the datapoints are examples of adversarial images at a given relative entropy. The orange line shows the test error (in %) of a mean field Boltzmann network over the whole clean MNIST test set, with the straight orange lines showing the reference test error of LeNet and a network with RBM pre training.

layer parameters are kept fixed during training and the remaining layers are trained in a standard discriminative fashion.

Let us establish the connection between better generative pre-training and increased adversarial resistance. We initialize the mean field Boltzmann network with parameters from various stages of the generative pre-training. As a measure for the generative training process we can use the relative entropy of the generated and target probability distribution, Eq. 1. To assess adversarial resistance we use an iterative gradient based (white box) method with $L_2$ constraints and record the mean squared error for the first image iteration that is misclassified. We average the mean squared error for 1000 randomly chosen images from the MNIST test set. We also calculate the test error for the whole clean MNIST test set for a given neural network. The results are presented in Fig. 3. We see that a smaller relative entropy in the pre-training phase correlates with a higher adversarial resistance. For comparison we apply the same procedure to a standard LeNet network as well as a neural network where the mean field Boltzmann network is based on an RBM with 64 visible and 128 hidden units trained in the same mean field approximation. Both LeNet and RBM pre-trained network do not show the strong adversarial resistance observed in the neural network using the mean field Boltzmann network as input layer. On the other hand we see that the classification error of the robust neural network on the clean test set, while above the LeNet and RBM error, is independent of the pre-training. We find no evidence for a strong correlation of classification error and adversarial resistance in our model.

One reason for the apparent robustness of neural networks are obfuscated gradients (Athalye et al. (2018)). Therefore we also evaluate the robust network using a gradient free (black box) method: the Boundary Method (Brendel et al. (2017)), implemented in Foolbox. For evaluation we use the network that showed the highest adversarial resistance to the iterated gradient method. The Bound-

Table 1: Comparison of our method with the ABS model. The entries in the table show the average $L_2$ norm and the robustness at a threshold of $\epsilon = 1.5$. Our method is tested over 1000 randomly selected images from the MNIST test set. Histograms of the $L_2$ norms of the different attacks can be found in appendix B. WB and BB indicates white box and black box attack methods, respectively. For the compound attack we select the lowest $L_2$ distance out of all four methods.

| model | Gradient Iterative (WB) | Boundary Method (BB) | Transfer Attack (BB) | Gaussian Noise Attack (BB) | Compound Attack |
|---|---|---|---|---|---|
| mean field Boltzmann network | 3.9 / 97 % | 3.46 / 85 % | 5.6 / 96 % | 10.6 / 98% | 2.73 / 82% |
| Analysis by Synthesis | 3.1 / 87 % | 2.6 / 83 % | 4.6 / 94 % | 10.9 / 98 % | 2.3 / 80% |

ary Method, using 1200 iterations, spherical_step=0.1 and otherwise default parameters achieves an average $L_2$ distance of

We have added two more attacks to our evaluation. The first attack is a transfer attack using a gradient based method on LeNet to obtain a perturbation. The amplitude of the perturbation is then increased until the network under test misclassifies the image. The second attack is a random Gaussian noise attack, where the variance of the added noise is increased until the network under test misclassifies the image. Using all four attacks we can construct a compound attack as suggested by Schott et al. (2018) by considering the attack with the smallest $L_2$ norm for each image. Combining all four attacks reduces the adversarial robustness below the robustness of any single attack.

Another reason for the apparent robustness of our model could be the very deep (20 layers) pre-processing stage leading to vanishing gradients. To investigate this further we remove the last, fully connected layer in our neural network and train a second, shallower network with the 32 x 6 x 6 dimensional activation data. The distilled network consists of 5 convolution layers: (8 x 8, stride 1, 256 filters), (1 x 1, stride 1, 256 filters), (3 x 3, stride 2, 128 filters, padding 1), (3 x 3, stride 2, 128 filters, padding 1), (3 x 3, stride 1, 32 filters, padding 1), with ReLU activations. Putting back the fully connected layer from the full network we can measure the adversarial resistance in the same way as earlier. We obtain an average $L_2$ distance of 3.35 and an error rate over the clean dataset of 1.35%, compared to an average $L_2$ distance of 3.9 for the original network. Although the distilled network is showing a loss of robustness, the average $L_2$ distance is still more than twice than the average $L_2$ distance for LeNet. This shows that some robustness can be transferred from the network structure used for learning to a relatively simple network and suggests that the robustness is not an artefact of a too deep network structure.

A comparison of our results with currently the most adversarially robust method, the Analysis By Synthesis (ABS) model of Schott et al. (2018), is shown in Tab. 1. For results from other methods see table 1 in Schott et al. (2018). Our model achieves comparable robustness results.

## 5 PROPERTIES OF MEAN FIELD BOLTZMANN NETWORKS

To understand the emerging adversarial robustness as a function of improved generative capability we examine the response of the mean field Boltzmann network around different input vectors. For one set of input vectors we will consider sample patterns from the training set, for the other set we select random binary patterns. We then add uniform random noise of amplitude $\epsilon$ to the input vectors and evaluate the different outputs of the network. The measure $\Delta_{r/t}$, presented in Fig. 4, is the $L_2$ norm of the average difference of clean and noisy output of a set of 1000 random and training set patterns respectively with 10000 random noise patterns for each input pattern. Training set patterns are much more resistant to noise than random patterns and the resistance increases as the Boltzmann machine training progresses, particularly for small amplitude noise.

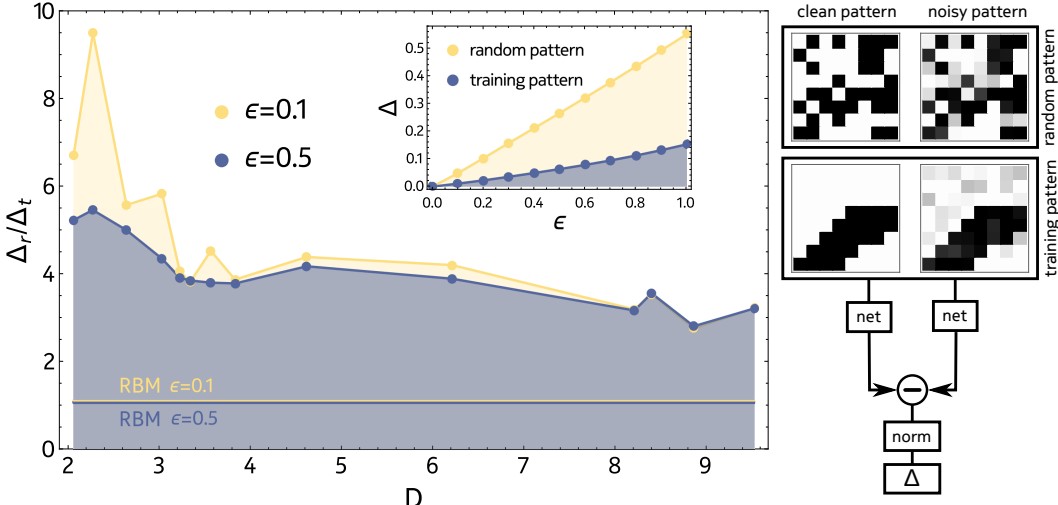

Figure 4: Response of a mean field Boltzmann network to noise at the input for different sets of input patterns. The inset shows how much adding random noise (uniformly distributed) of maximum amplitude $\epsilon$ to the input pattern changes the output pattern. The main plot shows the ratio of the differences for the random and training set patterns as a function of training progress of the Boltzmann machine as measured by the relative entropy.

## 6 DISCUSSION

We have shown that we can construct a feed forward neural network that shows strong adversarial resistance. This is achieved by incorporating a generatively pre-trained building block derived from the mean field description of a Boltzmann machine, which we called the mean field Boltzmann network. The resulting adversarial resistance strongly correlates with the effectiveness of the generative pre-training. We believe that the increased adversarial resistance can be traced to how the mean field Boltzmann network rejects noise around training example patches. The noise rejection does not happen for randomly selected examples. The rejection behaviour is an indication of the local non-linear behaviour of the mean field Boltzmann network. Compare this with the properties of a random matrix as feature extractor, where the restricted isometry property tells us that distances are preserved in the mapping with high probability. Too weak non-linearity of neural networks has been suggested as one reason for the existence of adversarial images and the increased non-linearity in our model correlates with increased adversarial resistance. On the level of individual image patches we can still find noise patterns that result in a large response when added to a training example, but they are much less likely. Take this together with the convolutional way that the mean field Boltzmann network is deployed. To get a large adversarial response from a given image region all overlapping image patches have to have a large response to the particular adversarial pattern. With the probability of large response supressed for one pattern, it becomes even smaller for all the overlapping pattern, resulting in strong adversarial resistance for the full neural network.

Our results, while encouraging, leave a number of questions open. First we have to ask ourselves if the success is due to the simplicity of the MNIST dataset. Will we be able to find typical image elements and model them with a Boltzmann machine for real image data? Second, while the adversarial images we find show a large deviation from the original image most do not resemble other image classes (for example images see appendix C). The algorithm still confidently places them in a particular class. We would prefer a behaviour where images unrecognisable by a human would result in a low confidence of the prediction.

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

## A   HIGHER ORDER TERMS IN THE MEAN FIELD EXPANSION

The third and fourth order term in the free energy becomes

$$F_3 = \sum_{\langle ijk \rangle} J_{ij} J_{jk} J_{ki} \left(1 - m_i^2\right) \left(1 - m_j^2\right) \left(1 - m_k^2\right)$$

$$+ \frac{8}{3} \sum_{\langle ij \rangle} J_{ij}^3 m_i \left(1 - m_i^2\right) m_j \left(1 - m_j^2\right),$$

$$F_4 = 2 \sum_{\langle ijkl \rangle} J_{ij} J_{jk} J_{kl} J_{li} \left(1 - m_i^2\right) \left(1 - m_j^2\right) \left(1 - m_k^2\right) \left(1 - m_l^2\right) +$$

$$+ 16 \sum_{\langle ijk \rangle} J_{ij}^2 J_{ik} J_{kj} m_i \left(1 - m_i^2\right) m_j \left(1 - m_j^2\right) \left(1 - m_k^2\right)$$

$$- \frac{2}{3} \sum_{\langle ij \rangle} J_{ij}^4 \left(1 - m_i^2\right) \left(1 - m_j^2\right) \left(1 + 3m_i^2 + 3m_j^2 - 15 m_i^2 m_j^2\right)$$

The third and fourth order terms for the mean field iteration are

$$R_i^{(3)} = -8 \sum_{\langle jk \rangle} J_{ij} J_{jk} J_{ki} m_i \left(1 - m_j^2\right) \left(1 - m_k^2\right)$$

$$+ \frac{2}{3} \sum_j J_{ij}^3 \left(1 - 3m_i^2\right) m_j \left(1 - m_j^2\right),$$

$$R_i^{(4)} = -16 \sum_{\langle jkl \rangle} J_{ij} J_{jk} J_{kl} J_{li} m_i \left(1 - m_j^2\right) \left(1 - m_k^2\right) \left(1 - m_l^2\right) +$$

$$- 32 \sum_{\langle jk \rangle} J_{kj}^2 J_{ij} J_{ki} m_i m_j \left(1 - m_j^2\right) m_k \left(1 - m_k^2\right)$$

$$+ 32 \sum_{\langle jk \rangle} J_{ij}^2 J_{kj} J_{ki} \left(1 - 3m_i^2\right) m_j \left(1 - m_j^2\right) \left(1 - m_k^2\right)$$

$$- \frac{16}{3} \sum_j J_{ij}^4 m_i \left(1 - m_j^2\right) \left[1 - 3m_i^2 + 3m_j^2 \left(5m_i^2 - 3\right)\right]$$

# B   HISTOGRAMS OF $L_2$ DISTANCES

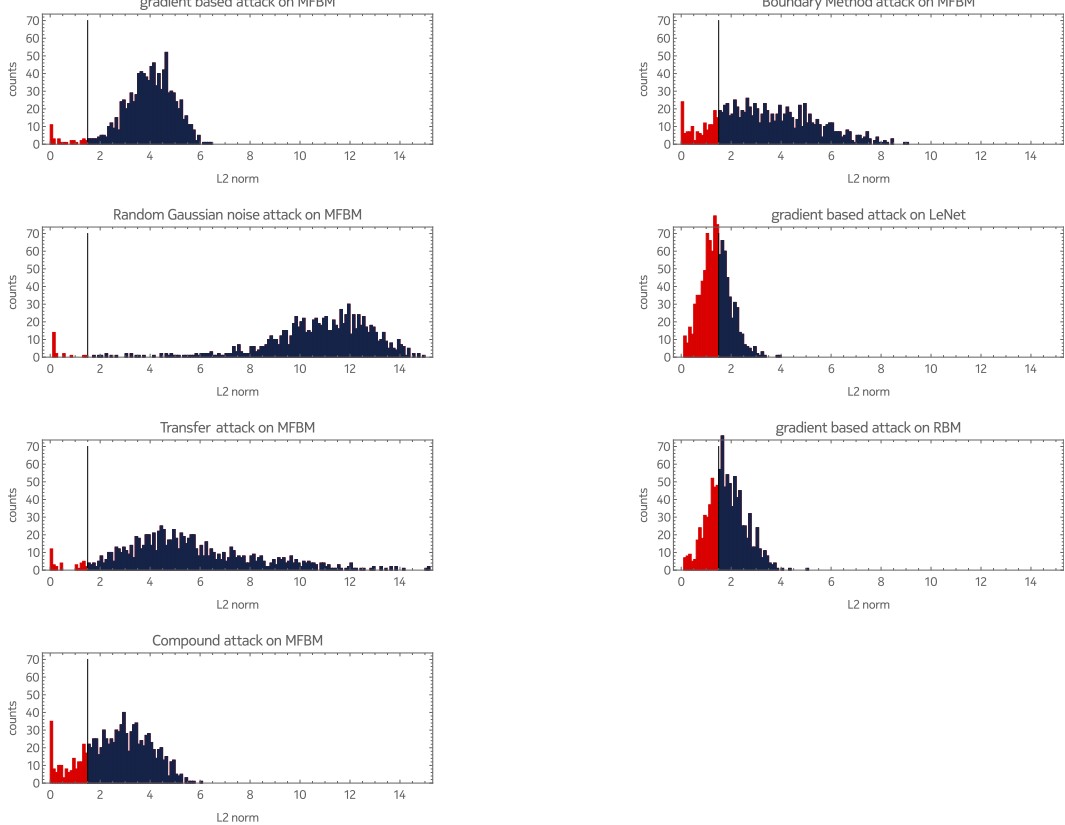

Figure 5: Histograms of the $L_2$ distances of adversarial images after various attacks for a set of 1000 images randomly selected from the MNIST training set. MFBM denotes the mean field Boltzmann machine approach developed in this article. The vertical line indicates the cut-off $\epsilon = 1.5$ used in Table 1.

## C  EXAMPLE ADVERSARIAL IMAGES

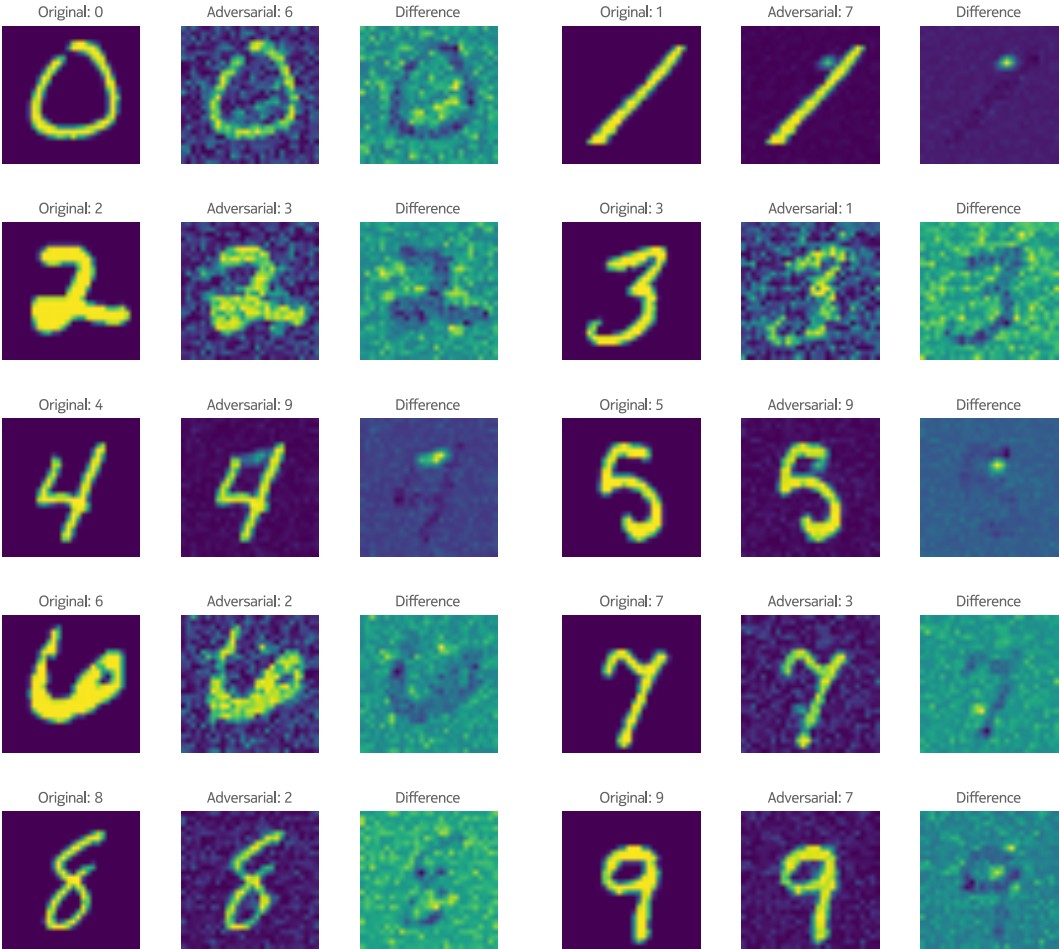

Figure 6: The first images of each class in the randomly selected images from the MNIST test set, their adversarial images obtained by the Boundary Method and the difference to the original image. The adversarial images are labeled by their misclassified labels. For some images the adversarial image clearly tends towards a human recognizable digit of another class (see 4, 5 and 9).

## D  MEAN FIELD TRAINING VS. EXACT SOLUTION

For a very small Boltzmann machine we can examine how the mean field training procedure used in the main part of the article compares to the exact solution. We select a Boltzmann machine of 16 units, which we divide into 5 visible units and 11 hidden units. We choose the connectivity to have bipartite connections between visible and hidden units and full connectivity between the hidden units. To examine the training behaviour we choose two example states out of 32 possible input states, (1, -1, -1, -1, 1) and (-1, -1, 1, 1, -1), to be realised with equal probability. For the training we use the mean-field derived relative entropy and its gradient from section 2. First, we can examine how the relative entropies from the mean field approach and the exact calculation compare during the training procedure. The results are presented in Fig. 7a. For most of the training the mean field derived relative entropy tracks the exact entropy very closely, only to be underestimated towards the end of the training. This is due to the underestimation of the partition function / free energy, resulting in the overestimation of probabilities. The probabilities are presented in Figs. 7b, 7c and their ratio in Fig. 7d. We see that although the probabilities are overestimated, their ratio is reproduced almost exactly.

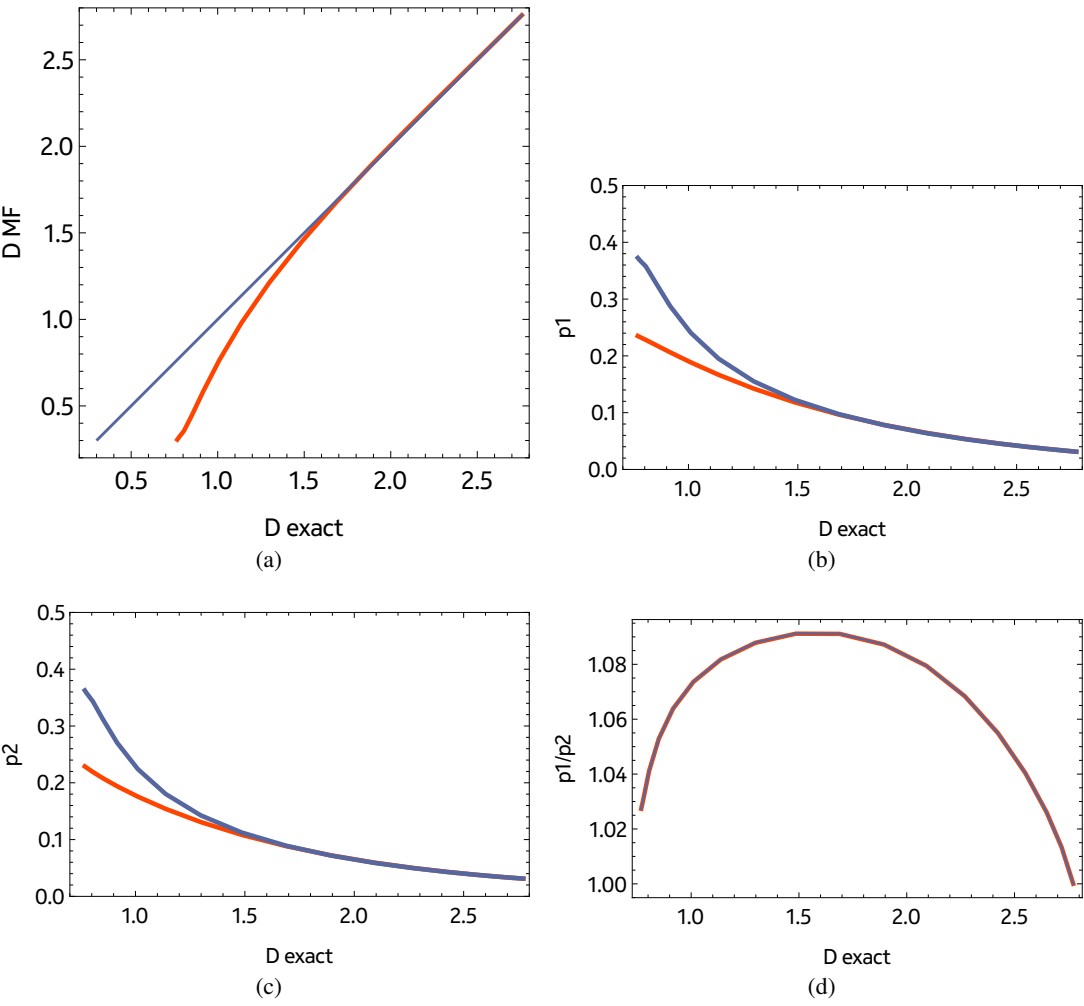

Figure 7: Comparison of an exact solution to the Boltzmann machine and the fourth order mean field method used in this article for different stages in the Boltzmann machine training. (a) Mean field derived relative entropy as a function of the exact relative entropy. The relative entropy does not reach zero during training. (b) Probability of the first example as a function of exact relative entropy. Red shows the mean field solution, blue the exact solution. (c) Probability of the second example as a function of exact relative entropy. (d) Ratio of the probabilities of first and second example.

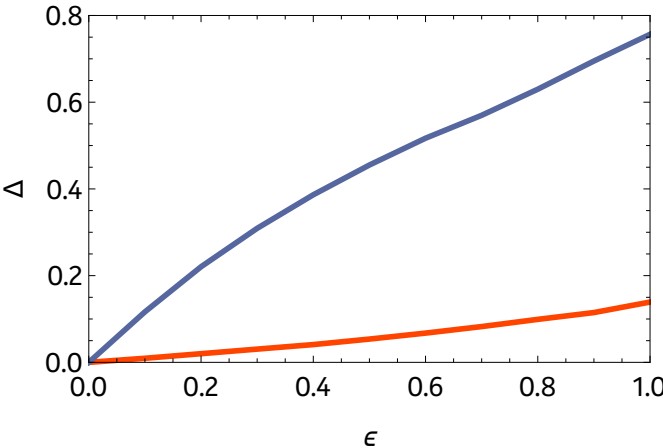

Figure 8: Response of a small mean field Boltzmann network to noise at the input for different sets of input patterns. The plot shows how much adding random noise (uniformly distributed) of maximum amplitude $\epsilon$ to the input pattern changes the output pattern. The red line shows the response averaged over the two training patterns, the blue line shows the response averaged over the remaining 30 patterns.

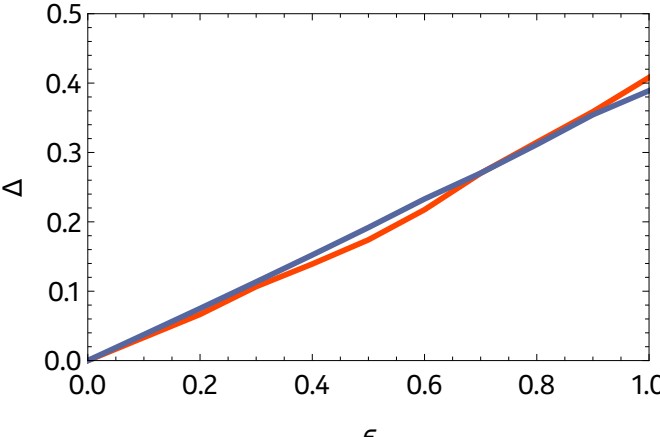

Figure 9: Response of a small restricted mean field Boltzmann network to noise at the input for different sets of input patterns. The plot shows how much adding random noise (uniformly distributed) of maximum amplitude $\epsilon$ to the input pattern changes the output pattern. The red line shows the response averaged over the two training patterns, the blue line shows the response averaged over the remaining 30 patterns.

We can now consider the noise rejection property for the small machine, using the same procedure as in the main article. For this small machine we can calculate the averages over the input states for all possible combination. To evaluate the noise rejection we used 1000 random noise patterns for each example. The results are shown in Fig. 8 and show again the same noise rejection behaviour for the example states. In contrast, if we use a simple RBM connected machine of the same size, we don't see any noise rejection behaviour (see Fig. 9).

