# OpenReview forum: "Improved resistance of neural networks to adversarial images through generative pre-training"
_ICLR.cc/2019/Conference_

### Official Review · AnonReviewer1 · 2018-11-02
**Fascinating work, and many questions**

**Rating:** 6
**Confidence:** 4

**Review:**

The recent work of Schott et al (which the authors compare results to) proposed the use of Bayes rule inversion as a more robust mechanism for classification under different types of adversarial attacks. The probabilities are approximated with variational autoencoders. During training the inference network is used, but during testing optimization based inference is carried out to compute loglikelihoods.

This paper focuses on the second part, with a different model. Specifically, it proposes a specific Boltzmann machine to be used as a first layer of neural networks for MNIST classification. This Boltzmann machine is pre-trained in two-stages using mean field inference of the binary latent variables and gradient estimation of the parameters.  This pre-trained model is then incorporated into the neural net for MNIST classification.  The existence of couplings J_h among the hidden units means that we have to carry out mean field inference over several iterations to compute the output activations of the model. This is basically analogous to the optimization-based inference proposed by Schott et al. (As a detail, this optimization can be expressed as computation over several layers of a neural net.)

The authors compare to the work of Schott for one type of attack. It would be nice to see more detailed experiments as done in Schott.

Questions:
1- Why not use a single Boltzmann machine with 128 fully connected latent variables? Could you add this experiment please.
2- Why is a two-stage pre-training (Figure 2) process needed? Why not just a single stage?
3- Is the key that you used only 679 patches containing 98% of occurrences in the dataset as the first stage? What if we vary this percentage? How sensitive are the results? Such experiments could be useful to understand better why your method appears to work well.
4- Could you please add the found J_h's to the appendix. This architecture reminds me of the good old MRFs for image denoising. Could it be that what we are seeing is the attack being denoised?

I am puzzled and looking forward to answers to the above questions. I don't yet understand what is the thing that makes this approach appear to work, or why you were able to drop the Bayes inference inversion altogether as done by Schott.

Thanks in advance. I will re-adjust the review rating following your reply.

---

> ### Author Response · Authors · 2018-11-26
> **Reply:**
>
> We thank the referee for their review and the summary of our results
>
> 1. We have included some more attacks on the most robust model (a transfer attack and a Gaussian random noise attack).
>
> 2.
>
> (a) We have evaluated the adversarial resistance when training a Boltzmann machine with 256 fully connected latent variables directly on the 8x8 patches. The version with only 128 hidden units was not able to reduce the relative entropy to the values of the larger, stacked machine. We find that the model without stacking is not able to increase the adversarial resistance. It is possible that we are unable to complete the training due to the approximations involved. For a small machine (16 units) of full hidden connectivity we can observe the noise rejection behaviour, as shown in appendix D.
>
> (b) We have trained a machine with the same connectivity as the stacked machine directly on the 8x8 patches. This training gives similar results to the training in stages.
>
> (c) From the result in (b) we conclude that the particular manner of the pre-training does not matter. Therefore also the choice of first training set (98% coverage or full coverage) does not influence adversarial resistance.
>
> (d) There are a total of 28800 parameters in the Boltzmann machine. We will gladly provide files with the trained weights and also fully trained neural networks on request.
>
> 3. We currently do not have a full explanation for the large adversarial resistance, but noise resistance must play a part in it. The very strong rejection of Gaussian noise and the observations in Fig. 4 point in this direction.

---

### Official Review · AnonReviewer2 · 2018-11-03
**Original Idea, Incomplete set of experiments**

**Rating:** 4
**Confidence:** 3

**Review:**

Authors propose a novel combination of RBM feature extractor and CNN classifiers to gain robustness toward adversarial attacks. They first train a small mean field boltzmann machine on 4x4 patches of MNIST, then combine 4 of these into a larger 8x8 feature extractor. Authors use the RBM 8x8 feature representation as a fixed convolutional layer and train a CNN on top of it. The intuition behind the idea is that since RBMs are generative, the RBM layer will act as a denoiser.

One question which is not addressed is the reason for only one RBM layer. In "Stacks of convolutional Restricted Boltzmann Machines for shift-invariant feature learning" by Norouzi et al, several RBM layers are trained greedily (same as here, only difference is contrastive loss vs mean field) and they achieve 0.67% error on MNIST. Attacking CRBMs is highly relevant and should be included as a baseline.

The only set of experiments are comparisons on first 500 MNIST test images. If the test set is not shuffled (by emphasis on first I assume not) these images are from training NIST (cleaner) set and may not include samples of all digits. Authors should clarify the justification behind experimenting only on 'first 500 test images'.

Furthermore, as authors discussed the iterative weight sharing which increases the depth can vanish the gradient toward input. Including at least one set of black box attacks is necessary to verify to what degree the vanishing gradient is the case here. The iterative architecture is similar to the routing in CapsNet (Hinton 2018) in terms of weight sharing between successive layers. Although their network was resilient toward white box attacks they suffered from black box attacks. The boundary method on MNIST could be  weaker than a black box attack.

---

> ### Author Response · Authors · 2018-11-26
> **Reply:**
>
> We thank the referee for their review.
>
> 1. We are not training Restricted Boltzmann Machines (RBMs), but Boltzmann machines where the hidden units can be fully connected.
>
> 2. The complete connectivity graph for our Boltzmann machine, as presented in Fig 2, can be interpreted as having two hidden layers. The graph has bipartite connectivity between the visible units and the first 128 hidden units and bipartite connectivity between the first 128 hidden units and the second 128 hidden units. We thank the referee for bringing the article [V] to our attention and we now have acknowledged the prior work properly in our introduction. We agree that it would be very instructive to evaluate the model in [V] for adversarial resistance, but we would argue that this evaluation is beyond the scope of this article.
>
> 3. Due to the complexity of the network compared to e.g. LeNet and the higher adversarial resistance the optimisation procedure to find adversarial images takes a long time, making it hard to evaluate 10000 images for all training stages and different attacks. We have now evaluated the adversarial resistance throughout the article for 1000 images randomly selected from the 10000 MNIST test images. This should avoid placing too much emphasis on the cleaner images in the beginning of the MNIST test set. Fig. 3 and other evaluations have been updated for the new test set.
>
> 4. To our knowledge the boundary method is the strongest black box attack. The succesful transfer attack on CapsNet is based on transfer of adversarial images from a different model (LeNet). We have implemented this attack and added it to our evaluation.
>
> [V] Norouzi, Mohammad Ranjbar, Mani Mori, Greg: Stacks of convolutional Restricted Boltzmann Machines for shift-invariant feature learning. 2009 IEEE Conference on Computer Vision and Pattern Recognition, 2735-2742 (2009).

---

### Official Review · AnonReviewer3 · 2018-11-05
**Interesting idea looking at generative models and its effect on defense against adversarial attacks. However, there are some key questions left unanswered.**

**Rating:** 4
**Confidence:** 4

**Review:**

This paper is clearly written and in an interesting domain. The question asked is whether or not pretrained mean-field RBMs can help in preventing adversarial attacks. However, there are some key issues with the paper that are not clear.

The first is regarding the structure of the paper. The authors combine two ideas, 1: the training of MF RBMs and 2: the the ability to prevent adversarial attacks. The combination of ideas is ok, however, it is unclear how novel or how good is the proposed MF training of the RBMs. It would make the paper much stronger if the authors perform quantitative + qualitative evaluation on the MF training of RBMs first.  Without doing so, it leaves the reader wondering why not simply a standard RBM trained using a standard method (e.g. contrastive divergence).

In a related note, using MF for training BMs have been proposed previously and found to not work due to various reasons:
see paragraph after equation 8 of the Deep BM paper: http://proceedings.mlr.press/v5/salakhutdinov09a/salakhutdinov09a.pdf

It would be very interesting to contrast the proposed method with other previously proposed MF based method, in particular using Free energy to approximate the expectation of the model without constraints.

It is also unclear how the calculation of relative entropy "D" was performed in figure 3. Obtaining the normalized marginal density in a BM is very challenging due to the partition function.

The second part of the paper associate good performance in preventing adversarial attacks with the possibility of denoising by the pretrained BM. This is a very good point, however the paper do not compare or contrast with existing methods. For example, it is curious to see how denoising Auto encoders would perform. In addition, it could be worthwhile to compare and benchmark on existing evaluations: https://arxiv.org/pdf/1802.06806.pdf

- The authors should make a distinction on what kinds of attack is considered: white box, black box or grey box. Defending against black box attacks is considerably easier than defending against white-box attacks.

In summary, the paper is interesting, however, more experiments could be added to concretely demonstrate the advantage  of the proposed MF BMs in increasing robustness against adversarial attacks.

---

> ### Author Response · Authors · 2018-11-26
> **Reply:**
>
> We thank the referee for finding our paper clearly written and in an interesting domain.
>
> 1. We are not training Restricted Boltzmann Machines (RBMs), but Boltzmann machines where the hidden units can be fully connected. We have shown in our experiments (see Fig. 3) that using a single RBM as a pre-processor will not result in increased adversarial resistance. A comparison of mean field training vs. constrastive divergence for RBMs has been made by [III,IV]. Nevertheless, we have added a comparison between our method and the exact solution for a small Boltzmann machine of 16 units in the appendix. While we don't claim that this explains the performance of the method under all circumstances, it gives an indication of how well it works. We agree that a more thorough study of the training method would be desirable but in this article we are concentrating on reporting the results on increased adversarial resistance. We believe that these results should be interesting to the community even if some doubts about the training procedure remain. Lastly, we want to obtain a differentiable building block that can be used in standard neural nets. The unrolling of the mean field iterations (see Fig. 1) provides a straightforward to achieve this. Propagating a gradient through a sampling based building block, while possible, would be considerably harder.
>
> 2. In [III,IV] the authors have succesfully used a mean field approximation beyond the trivial first order to train RBMs. In [II] a first order approximation is used. In our approach we use the approximation up to fourth order in the coupling J.
>
> 3. The approach of the authors in [III,IV] and also our approach is based on free energy. The systematic expansion of the free energy in the coupling constant forms the basis of our approach.
>
> 4. D is calculated as the relative entropy over a batch of 10000 examples. The reference probabilities are taken to be uniform and the model probabilities are calculated according to the procedure outlined in section 2. Due to the approximations involved it is possible that the sum of the model probabilities exceeds one. In this case we rescale the unclamped free energy to limit the total probability to one. We have added a note to that effect to the article.
>
> 5. We have included a comparison to the, to our knowledge, currently most adversarially resistant model on MNIST. Since our results are derived for MNIST we can only compare to methods in the literature that are evaluated on MNIST. We acknowledge that other methods perform very well on more sophisticated tasks and have added a reference.
>
> 6. While we have not explicitely labelled white box and black box attacks we are using a strong white box attack (gradient based) and the, to our knowledge, strongest black box attack (boundary method). We have added the labels in the text and Table 1 to make this more clear.
>
> 7. We have added some more attacks on the most robust model. We completely agree that more experiments on other datasets are needed to show the universality of the method. Due to the computational complexity this will need to remain a topic for a follow up article.
>
> [I] Seyed-Mohsen Moosavi-Dezfooli, Ashish Shrivastava, Oncel Tuzel: Divide, Denoise, and Defend against Adversarial Attacks. arXiv 1802.06806 (2018).
>
> [II] Ruslan Salakhutdinov, Geoffrey Hinton: Deep Boltzmann Machines. Proceedings of the Twelfth International Conference on Artificial Intelligence and Statistics, 448-455, (2009)
>
> [III] Paweł Budzianowski. Training Restricted Boltzmann Machines Using High-Temperature Expansions. Master’s thesis, University of Cambridge, 2016.
>
> [IV] Marylou Gabrié, Eric W. Tramel, and Florent Krzakala. Training restricted boltzmann machines via the Thouless-Anderson-Palmer free energy. CoRR, abs/1506.02914, 2015.

---

> > ### Comment · AnonReviewer3 · 2018-12-11
> > **update scores**
> >
> > In light of the clarifications and more details from the authors,  I would like to adjust the score upwards to *5*. In general, it is not surprising that better generative models will lead to better defenses against attacks. The more interesting question is *how much* resistant might be a generatively pretrained networks to adversarial images. I think the experimental section could still use more comparisons with other existing defense methods. For example, methods from R1 could be ported to MNIST to make the experiment baselines much stronger.
> >
> > - In addition, a suggestion is perhaps to try MCMC-based algorithm such as Annealed importance sampling to calculate D, as it might be an alternative way and reduce approximation error.
> >
> > [R1] https://www.kaggle.com/c/nips-2017-defense-against-adversarial-attack/data

---

> > > ### Author Response · Authors · 2018-12-12
> > > **Quantitative Analysis**
> > >
> > > We give a quantitative analysis of the effectiveness of our method in Tab. 1 and more details in the histograms in Fig. 5. We give a comparison to the, to our knowledge, strongest defense against adversarial attacks on MNIST [VI]. In the same article, [IV], some more model evaluations can be found. This should be sufficient to evaluate the strengths of our model and make it easy enough for other authors to compare to our method, if they choose to do so.
> > >
> > > [VI] L. Schott, J. Rauber, W. Brendel, and M. Bethge. Towards the first adversarially robust neural
> > > network model on mnist. 2018.

---

### Author Response · Authors · 2018-11-26
**New version of article uploaded**

The values for the thresholded success rates in table 1 are increased, since in the original submission we accidentally divided the L2 distances by a factor of 4 instead of a factor 2 while translating from our working interval [-1,1] to the measurement interval [0,1].

We have improved the efficiency of the boundary attack by parameter tuning. In the original submission the boundary attack was used with default parameters. After adjusting spherical_step to 0.1 we achieved a much smaller L2 distance.

We have added a figure in the appendix showing the histograms of L2 distances for various attacks.

We have added example adversarial images in the appendix.

We have added a short study comparing our mean field method to the exact solution for a very small Boltzmann machine of 16 units in the appendix.

---

### Meta-Review · Area_Chair1 · 2018-12-13
**No reviewer has championed accepting this paper**

**Confidence:** 5
**Recommendation:** Reject

**Metareview:**

No reviewer has made a strong case for accepting this paper or championed it so I am recommending rejecting it. The unfavorable reviewers, although they mention real issues, have not highlighted some of the most important barriers to accepting this work.

One major, but not necessarily dispositive, concern is that the paper only presents results on MNIST. However, even if we put aside this concern, there are several issues with the motivation and approach of this paper. If this technique is actually good at improving the model outside the clean image distribution, then the paper should show that and not just L2 worst case perturbations. To quote the intro of the paper: "How can deep learning systems successfully generalise and at the same time be extremely vulnerable to minute changes in the input?" The answer is: they don't generalize and this work does not show us improved generalization. Even a small amount of test error in the data distribution suggests that the closest test error to a given point will often be quite close to the starting point, although this is easier to see with linear models. The best way to fix this work would be to study (average case) error on noisy distributions (as in the concurrent submission https://openreview.net/forum?id=S1xoy3CcYX ).